# Loss of adult skeletal muscle stem cells drives age-related neuromuscular junction degeneration

**Wenxuan Liu[1,2], Alanna Klose[1], Sophie Forman[3], Nicole D Paris[1], Lan Wei-LaPierre[4], Mariela Cortés-Lopéz[5], Aidi Tan[6,7], Morgan Flaherty[2], Pedro Miura[5], Robert T Dirksen[4], Joe V Chakkalakal[1,8,9]***

[1]Department of Orthopaedics and Rehabilitation, Center for Musculoskeletal Research, University of Rochester Medical Center, Rochester, United States; [2]Department of Biomedical Genetics, University of Rochester Medical Center, Rochester, United States; [3]Department of Biology, University of Rochester, Rochester, United States; [4]Department of Pharmacology and Physiology, University of Rochester Medical Center, Rochester, United States; [5]Department of Biology, University of Nevada, Reno, United States; [6]Bioinformatics Division and Center for Synthetic and Systems Biology, Tsinghua University, Beijing, China; [7]TNLIST/ Department of Automation, Tsinghua University, Beijing, China; [8]Stem Cell and Regenerative Medicine Institute, University of Rochester Medical Center, Rochester, United States; [9]The Rochester Aging Research Center, University of Rochester Medical Center, Rochester, United States

**\*For correspondence:**
joe_chakkalakal@urmc.rochester.edu

**Competing interests:** The authors declare that no competing interests exist.

**Abstract** Neuromuscular junction degeneration is a prominent aspect of sarcopenia, the age-associated loss of skeletal muscle integrity. Previously, we showed that muscle stem cells activate and contribute to mouse neuromuscular junction regeneration in response to denervation (Liu et al., 2015). Here, we examined gene expression profiles and neuromuscular junction integrity in aged mouse muscles, and unexpectedly found limited denervation despite a high level of degenerated neuromuscular junctions. Instead, degenerated neuromuscular junctions were associated with reduced contribution from muscle stem cells. Indeed, muscle stem cell depletion was sufficient to induce neuromuscular junction degeneration at a younger age. Conversely, prevention of muscle stem cell and derived myonuclei loss was associated with attenuation of age-related neuromuscular junction degeneration, muscle atrophy, and the promotion of aged muscle force generation. Our observations demonstrate that deficiencies in muscle stem cell fate and post-synaptic myogenesis provide a cellular basis for age-related neuromuscular junction degeneration and associated skeletal muscle decline.

## Introduction

Age-related neuromuscular junction (NMJ, synapse between motor neuron and skeletal muscle cells) declines in rodents and humans manifest as altered pre-synaptic nerve terminal and abnormal post-synaptic morphology indicative of degeneration (*Gonzalez-Freire et al., 2014*; *Oda, 1984*; *Wokke et al., 1990*). Although these disruptions are often associated with skeletal muscle dysfunction and disease, whether age-associated NMJ remodeling primarily reflects deterioration of target muscle cells or the peripheral nervous system is unknown (*Kang and Lichtman, 2013*; *Banker et al., 1983*; *Li et al., 2011*).

It is well established that adult skeletal muscle possesses a remarkable capacity for regeneration endowed by a population of adult resident stem cells, satellite cells (SCs) (*Relaix and Zammit, 2012*). Through depletion studies, SCs, identified by the expression of the paired box transcription factor Pax7, are deemed to be the principal source of myonuclei for myofiber regeneration and repair (*Liu et al., 2015*; *Relaix and Zammit, 2012*). In our previous study, we examined SC fate during NMJ denervation and reinnervation, and found that SCs contribute to NMJ regeneration (*Liu et al., 2015*). Recently contributions of SCs and derived progenitors to myofibers have been observed during sedentary aging (*Keefe et al., 2015*). Extensive studies have shown that SC number and function decline with age; however, whether this loss is connected to age-associated NMJ degeneration is unknown (*Chakkalakal et al., 2012*; *Sousa-Victor et al., 2015*). Specifically, whether SCs contribute to the maintenance of aging NMJs, and if age-related SC loss primarily drives NMJ deterioration has not been tested.

In this study, we test to what extent age-related NMJ deterioration is connected to loss of SCs using fate tracking, depletion, and rescue strategies. Our results reveal that loss of SC-derived contribution to post-synaptic myofiber maintenance, as opposed to denervation, is more closely associated with age-related NMJ deterioration and skeletal muscle atrophy.

## Results

### Complete denervation is not a prominent feature of aged skeletal muscles

Aging is associated with an increased proportion of NMJs that display degenerated pre/post-synaptic morphologies in both tibialis anterior (TA) and diaphragm muscles (*Figure 1A–E*) (*Gonzalez-Freire et al., 2014*; *Valdez et al., 2010*, *2012*). Although we observed a high proportion of degenerated NMJs in aged muscles, very few were completely denervated (*Figure 1B and D*). To further examine the extent to which aged skeletal muscles are denervated we performed genome-wide RNA-seq analysis on synaptic (Syn) and extra-synaptic (ExS) regions of adult (6-month-old) and aged (24-month-old) diaphragms (*Figure 1F*). To confirm the fidelity of our samples, we conducted gene set enrichment analysis (GSEA) between 6M-Syn and 6M-ExS samples using gene sets derived from the Gene Ontology Molecular Function (GOMF). As expected, gene sets associated with acetylcholine receptor (AChR) activity and neurotransmission were highly enriched in 6M-Syn samples (*Figure 1—figure supplement 1A and B*) (*Chakkalakal and Jasmin, 2003*; *Merlie and Sanes, 1985*; *Hippenmeyer et al., 2007*; *Kishi et al., 2005*). Subsequently in aged muscles we examined the expression levels of synaptic genes, which are elevated in a model of complete denervation and reinnervation leading to degenerated NMJ morphology (*Figure 1—figure supplement 2*) (*Liu et al., 2015*; *Kishi et al., 2005*; *Adams et al., 1995*; *Lin et al., 2001*). Although we observed previously described alterations in mitochondrial metabolism and atrophy-related genes in 24M-ExS and 24M-Syn samples, consistent with low levels of complete denervation we did not observe elevation or extra-synaptic induction of synaptic gene expression (*Figure 1G*, *Figure 1—figure supplement 1C–H*) (*Hippenmeyer et al., 2007*; *Kishi et al., 2005*; *Ibebunjo et al., 2013*). However, comparison of 6M-Syn and 24M-Syn samples revealed age-related declines in synapse-specific gene expression, indicating a possible loss of post-synaptic myonuclei, the source of synapse-specific mRNAs at NMJs (*Figure 1G*, *Figure 1—figure supplement 1C–H*).

Although other non-myogenic cell types such as Schwann cells are present at NMJs on nerve terminals, they are completely segregated from the myofiber by an extensive basal lamina network. Myonuclei clustered beneath the post-synaptic membrane of the NMJ within the myofiber are transcriptionally distinct from adjacent myonuclei and specialized for the expression of genes critical for NMJ function (*Chakkalakal and Jasmin, 2003*; *Schaeffer et al., 2001*). Reduction in the size of post-synaptic myonuclear clusters is associated with impaired NMJ development, ineffective regeneration after nerve insult, and neuromuscular diseases (*Liu et al., 2015*; *Zhang et al., 2007*; *Méjat et al., 2009*). The size of NMJs, based on post-synaptic apparatus volume and cumulative AChR intensity, positively correlated with the size of post-synaptic myonuclear clusters (*Figure 2A and B*, *Figure 2—figure supplement 1*). Notably, the majority of degenerated NMJs had few post-synaptic myonuclei (*Figure 2C*). We observed that the fraction of aged NMJ post-synaptic myonuclear clusters with greater than 4 myonuclei, in comparison to adult, was significantly lower

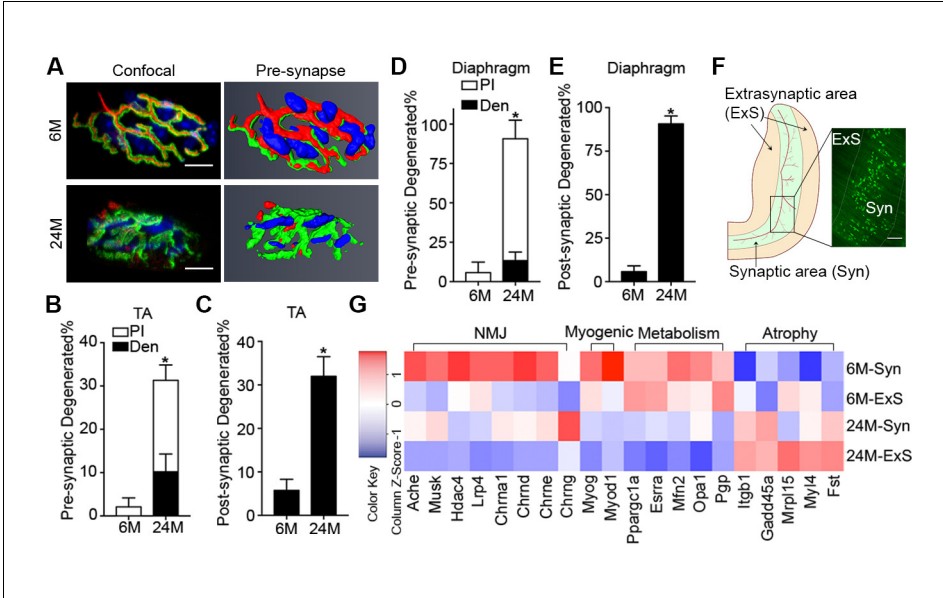

**Figure 1.** Aging of skeletal muscles is associated with NMJ degeneration without gross denervation. (A) Confocal IF images and 3-D Amira-based reconstructions (viewed from the pre-synaptic side) of adult (6M) and aged (24M) C57BL6/J NMJs stained for post-synaptic AChRs (α-bungarotoxin, BTX; green), nerve terminal markers (SV2, Syt-2, neurofilament; red) and nuclei (DAPI, blue). Scale bar = 10 μm. (B) Quantification of pre-synaptic degenerated NMJs, where pre-synaptic nerve terminals are still present but partially missing (partially innervated, PI), or fully devoid (completely denervated, Den), in adult and aged Tibialis Anterior (TA) muscles. (C) Quantification of post-synaptic degenerated NMJs, where the AChR-enriched area loses the elaborate branch morphology, in adult and aged TA muscles. (D–E) Quantification of (D) pre-synaptic degenerated or (E) post-synaptic degenerated NMJ percentage in adult and aged diaphragms. (F) Schematic of a diaphragm demonstrating the central NMJ clusters in diaphragm. Red lines indicate nerve terminals; inset is a representative image of a diaphragm stained with BTX (shown in green). Scale bar = 200 μm. (G) Heat map of FPKM values from adult and aged synaptic (Syn) and extra-synaptic (ExS) diaphragms. Red indicates increased expression, and blue indicates decreased expression. $n = 4$ mice per group, >20 NMJs/mouse (B–C); $n = 3$ mice per group, 20 NMJs/mouse (D–E); $n = 2$ (adult) or 3 (aged) mice (G). *$p < 0.05$ compared to adult, unpaired Student's $t$-tests.

The following figure supplements are available for figure 1:

**Figure supplement 1.** Synaptic specific gene enrichment declines with age.

**Figure supplement 2.** Aged skeletal muscle does not display transcriptional changes characteristic of denervation.

(*Figure 2D–F*). Therefore, loss of post-synaptic myonuclei is a predominant feature of aged degenerated NMJs.

## SCs contribute to adult and middle-aged, but not aged NMJs

Given that the size of post-synaptic myonuclear clusters is reduced with age and aging is associated with loss of SCs, the principal source of myonuclei for skeletal muscle regeneration (*Sousa-Victor et al., 2015*), we examined whether SCs are preferentially lost with age from Syn or ExS regions. To assess SC number with regard to their location relative to NMJs, we performed previously characterized flow cytometric analysis to prospectively isolate Syn and ExS SCs (CD31-/CD45-/Sca1-/ITGA7+/VCAM+) from 6, 12, 18 and 24-month-old C57BL/6J mice (*Chakkalakal et al., 2012*; *Maesner et al., 2016*). Regardless of their source (Syn vs ExS regions), SC frequency significantly declined at 18 and 24 months (*Figure 2—figure supplement 2A and B*). In vitro fate analysis of sorted Syn and ExS SCs confirmed their purity and capacity for myogenic differentiation (*Figure 2—figure supplement 2C*).

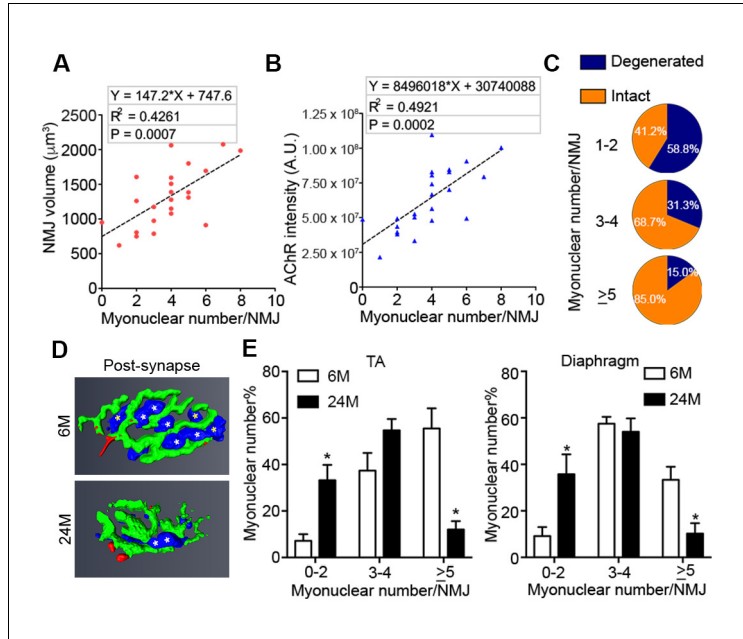

**Figure 2.** Aged-associated NMJ degeneration is accompanied by reduction in the size of post-synaptic myonuclear clusters. (**A–B**) Linear regression analysis demonstrates positive correlation between (**A**) NMJ volume or (**B**) AChR cumulative intensity (both quantified by Amira using confocal-fluorescent images of individual NMJs) and the post-synaptic myonuclear number. (**C**) Percentage of pre/post-synaptic degenerated or intact NMJs in groups with different size of post-synaptic myonuclear cluster. (**D**) 3-D Amira-based reconstructions (viewed from the post-synaptic side) of adult and aged C57BL6/J NMJs stained for post-synaptic AChRs (green), nerve terminal markers (red) and nuclei (blue). Post-synaptic myonuclei are indicated with white asterisks and pre-synaptic nuclei with yellow asterisks. (**E**) Percentage distribution of NMJs based on number of post-synaptic myonuclei in TA and diaphragm.

The following figure supplements are available for figure 2:

**Figure supplement 1.** NMJ size, based on post-synaptic apparatus volume and AChR intensity, positively correlates with the size of post-synaptic myonuclear clusters.

**Figure supplement 2.** Age-related SC decline occurs to a similar extent between synaptic and extra-synaptic diaphragm regions.

---

To examine SC and derived progenitor contribution to aging NMJs, we utilized a $Pax7^{CreER/+}$; $Rosa26^{mTmG/+}$ (P7mTmG) transgenic mouse line. The P7mTmG mouse ubiquitously expresses a loxP-flanked membrane tomato, red fluorescent reporter that undergoes tamoxifen-mediated excision to indelibly label Pax7+ SCs and derived cells with membrane GFP (*Liu et al., 2015*; *Keefe et al., 2015*). To examine SC-derived contribution at distinct ages, tamoxifen was administered at 6, 12, 18 and 24 months of age (*Figure 3A*). Single myofibers were isolated from P7mTmG extensor digitorum longus (EDL) muscles 6 weeks after tamoxifen administration, a relatively short time frame (*Keefe et al., 2015*), and processed for GFP and NMJ detection. At 6 and 12 months we found GFP-labeled SC derived contribution at myofiber ends where sarcomere addition can occur (*Williams and Goldspink, 1971*; *Zhang and McLennan, 1995*; *Wang et al., 2010*), and/or middle portions of myofibers where NMJs are located (*Liu et al., 2015*) (*Figure 3A,B and D*). To determine if NMJ-associated mGFP labeling comprised SC-derived contribution to post-synaptic myonuclei we utilized a $Pax7^{CreER/+}$; $Rosa26^{nTnG/+}$ (P7nTnG) transgenic mouse line. The P7nTnG mouse ubiquitously expresses a loxP-flanked nuclear Td-tomato, red fluorescent reporter that undergoes tamoxifen-mediated excision to indelibly label Pax7+ SCs and derived cells with nuclear GFP (*Prigge et al., 2013*). Immediately after tamoxifen administration to 4.5-month-old P7nTnG mice only Pax7+ SCs were GFP-labeled (*Figure 3—figure supplement 1A–C*). Consistent with SCs as a cellular source, 6

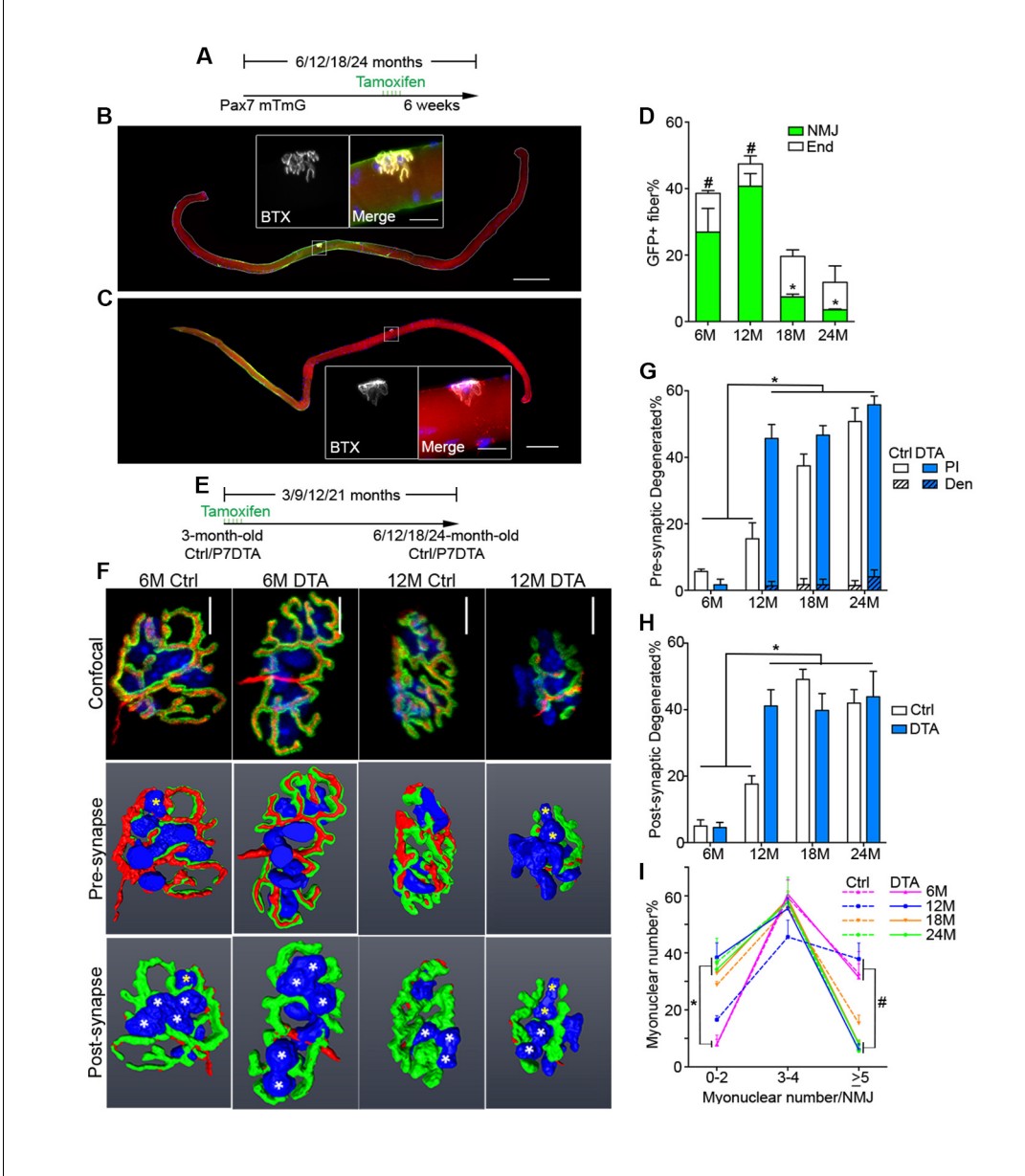

**Figure 3.** Contribution of SCs to NMJs is lost in aged muscle and SC depletion accelerates age-associated NMJ degeneration. (**A**) Scheme demonstrating time of tamoxifen treatment and harvest of tissue for P7mTmG mice. (**B**–**C**) Representative images of single isolated P7mTmG EDL myofibers from (**B**) 12-month-old mice, where mGFP is in middle portions where the NMJ is located or (**C**) 18-month-old mice, where mGFP is located at the end of the fiber. Magnified inset images show NMJs. Scale bar for myofibers = 200 μm, for inset = 25 μm. (**D**) Quantification of mGFP+ fiber percentage and distribution based on mGFP location (at the end of the fiber or in the vicinity of the NMJ). *p<0.05 for NMJ-associated mGFP compared to 6M/12M groups, #p<0.05 for mGFP at the end compared to NMJ-associated mGFP at that age group, two-way ANOVA/Sidak multiple comparison test. (**E**) Scheme demonstrating time of tamoxifen treatment and harvest of tissue for Ctrl or P7DTA mice. (**F**) Confocal IF images and 3-D Amira-based reconstructions (viewed from the pre-synaptic or post-synaptic side) of Ctrl and P7DTA NMJs stained for post-synaptic AChRs (green), nerve terminal markers (red) and nuclei (blue). Post-synaptic myonuclei are indicated with white asterisks, and representative pre-synaptic nuclei indicated with yellow asterisks. Scale bar = 10 μm. (**G**–**H**) Quantification of (**G**) pre-synaptic degenerated, including partially innervated (PI) or completely denervated (Den), or (**H**) post-synaptic degenerated NMJ percentage of Ctrl and P7DTA TA muscles. *p<0.05, 6M Ctrl/P7DTA, 12M Ctrl vs. 12M P7DTA, 18/24M Ctrl/P7DTA, two-way ANOVA/Sidak multiple comparison test. (**I**) Percentage distribution of TA NMJs based on number of post-synaptic myonuclei. *p<0.05, 6M Ctrl/P7DTA vs. 12M/18M P7DTA, 24M Ctrl/P7DTA; #p<0.05, 6M Ctrl/P7DTA, 12M Ctrl vs. 12M/18M P7DTA,

*Figure 3 continued on next page*

*Figure 3 continued*

24M Ctrl/P7DTA; two-way ANOVA/Sidak multiple comparison test. *n* = 3 mice per group, ≥50 myofibers examined per mouse (D); *n* = 3 (18M Ctrl, 24M Ctrl, 24M P7DTA); 4 (6M Ctrl, 6M P7DTA, 18M P7DTA) or 5 (12M Ctrl, 12M P7DTA) mice, >20 NMJs/mouse (**G–I**).

The following figure supplements are available for figure 3:

**Figure supplement 1.** Contribution of SCs to NMJs in adult muscles.

**Figure supplement 2.** Pax7+ SCs are persistently depleted throughout the lifespan in P7DTA muscles.

**Figure supplement 3.** Post-synaptic degeneration of NMJs occurs prior to pre-synaptic degeneration with aging.

**Figure supplement 4.** SCs are depleted in P7DTA diaphragms, accompanied by reduction in the size of post-synaptic myonuclear clusters.

**Figure supplement 5.** Age-related myofiber atrophy is observed in middle-aged P7DTA muscles.

weeks after tamoxifen administration GFP-labeled post-synaptic myonuclei were observed at NMJs (*Figure 3—figure supplement 1D–E*). In accordance with the timing of SC loss during aging (*Sousa-Victor et al., 2015*) and a potential role for SCs in the lifelong maintenance of NMJs, we observed a pronounced decline in GFP-labeled SC-derived contribution in the vicinity of NMJs in 18 and 24-month-old mice (*Figure 3B–D*).

## SC depletion accelerates the onset of age-associated NMJ degeneration

To test whether SC contribution is required for the maintenance of NMJs in aging mice, we employed previously characterized *Pax7*$^{CreER/+}$; *Rosa26*$^{DTA/+}$ (P7DTA) and *Pax7*$^{+/+}$; *Rosa26*$^{DTA/+}$ (Ctrl) mice (*Liu et al., 2015*). These mice enable tamoxifen-mediated expression of diphtheria tox-in-A (DTA) to deplete Pax7+ SCs, which is sufficient to prevent skeletal muscle regeneration (*Relaix and Zammit, 2012*; *Murphy et al., 2011*). We observed persistent depletion of Pax7+ SCs when tamoxifen was administered at 3 months, determined by Pax7 immunofluorescence analysis (*Figure 3E*, *Figure 3—figure supplement 2*). We identified an increased proportion of pre/post-synaptic degenerated NMJs in Ctrl skeletal muscles at 18 and 24 months (*Figure 3F–H*) (*Valdez et al., 2010*). Examination of P7DTA NMJs revealed induction of degenerated NMJs at 12 months, indicating that SC depletion markedly accelerated sarcopenia-related NMJ deterioration (*Figure 3F–H*). Prior to significant induction of partial innervation, a significant increase in the proportion of NMJs with 0–2 post-synaptic myonuclei was observed at 12 months compared to 6 month-old muscles (*Figure 3—figure supplement 3*). Indeed, the exacerbated deterioration of NMJs at 12 months, as a result of SC depletion, was also associated with a further decline in the size of post-synaptic myonuclear clusters (*Figure 3F and I*, *Figure 3—figure supplement 4C*).

Given that SC depletion accelerates the onset of age-related NMJ deterioration, and NMJ disruption is associated with myofiber decline, we examined the effects of SC depletion on age-related myofiber type transition and atrophy. Although we observed a reduction in the proportion of fast type IIX myofibers with age, the onset or severity of this phenotype was not altered upon SC depletion (*Figure 3—figure supplement 5A and B*). The examination of single myofibers isolated from EDLs, fixed prior to dissection, revealed significant reduction in cross-sectional area of myofibers upon SC depletion at 12 months (*Figure 3—figure supplement 5C and D*). Collectively, these data demonstrate that SC depletion is sufficient to induce early onset of sarcopenia phenotypes, specifically NMJ deterioration and myofiber atrophy without evidence of gross denervation.

## Aged NMJ and skeletal muscle integrity is improved upon Spry1 overexpression in SCs

The suppression of *Spry1*, a downstream target and negative-feedback modulator of receptor tyrosine kinase (RTK)/fibroblast growth factor (FGF) signaling (*Kim and Bar-Sagi, 2004*), has been linked

to SC pool loss in aged muscles (*Chakkalakal et al., 2012*; *Bigot et al., 2015*). Moreover, the forced expression of Spry1 in SCs attenuates age-associated SC decline (*Chakkalakal et al., 2012*; *Bigot et al., 2015*). To examine whether attenuation of age-associated NMJ deterioration and skeletal muscle declines can be associated with preservation of the SC pool, we utilized $Pax7^{CreER/+}$; $CAG^{Spry1/+}$ (Spry1OX) mice, which enable SC-specific Spry1 overexpression, and $Pax7^{+/+}$; $CAG^{Spry1/+}$ (Ctrl) mice (*Chakkalakal et al., 2012*; *Yang et al., 2008*). Flow cytometric analysis confirmed that aged muscles from Spry1OX mice with tamoxifen administered at 12 months had significantly more SCs (*Figure 4A*, *Figure 4—figure supplement 1*) (*Chakkalakal et al., 2012*). Examination of NMJ integrity revealed that age-related NMJ deterioration was attenuated in Spry1OX muscles, demonstrated by a reduction in the proportion of pre/post-synaptic degenerated NMJs (*Figure 4B–D*). Also, an increase in the size of post-synaptic myonuclear clusters was observed in aged Spry1OX mice (*Figure 4B and E*). Since we observed improvements in NMJ integrity and robust increases in the size of post-synaptic myonuclear clusters, we wondered if these findings were associated with improvements in skeletal muscle function. Therefore, we analyzed time or distance to exhaustion with treadmill exercise as an initial readout of muscle performance. Aged Spry1OX mice ran a significantly longer time/distance than Ctrl mice (*Figure 4F and G*). Next, we specifically examined skeletal muscle function through testing EDL *ex vivo* tetanic force generation capacity. Significant increases were detected in peak absolute force generated by aged Spry1OX muscles in comparison to Ctrl muscles upon stimulation at frequencies of 100, 125 and 150 Hz (*Figure 4H*, *Figure 4—figure supplement 2A and B*). However, no significant difference was observed when force was normalized to physiological cross-sectional area (specific force), which accounts for modifications in muscle girth (*Figure 4—figure supplement 2C*). Indeed, we observed increased EDL myofiber cross-sectional area without significant changes in myofiber type composition or number in aged Spry1OX mice (*Figure 4—figure supplement 2D–F*). Therefore, preventing age-related SC loss, through stimulation of Spry1 expression in SCs without direct peripheral nerve manipulation, was associated with the prevention of post-synaptic myonuclear loss, myofiber atrophy, and force generation capacity decline in aged skeletal muscles.

## Discussion

In this report, we find marginal denervation in aged skeletal muscles, and that loss of SC contribution drives age-related NMJ degeneration. Specifically, SCs are a source of post-synaptic myonuclei. With age SC-derived contribution of myonuclei to the post-synaptic region is compromised, which accordingly links to age-related NMJ degeneration. Remarkably, SC depletion was sufficient to accelerate the onset of sarcopenia-related declines in post-synaptic myonuclei, NMJ integrity, and myofiber size. In contrast, attenuating SC loss during aging through SC-specific Spry1 overexpression was associated with improved NMJ integrity and force generation capacity in aged muscles (*Figure 4—figure supplement 3*).

Intuitively, denervation is considered to be a mechanism that contributes to age-related skeletal muscle decline (*Gonzalez-Freire et al., 2014*). Although denervation is sufficient to trigger extensive skeletal muscle atrophy and dysfunction, the proportion of completely denervated aged NMJs is modest (*Figure 1B and D*) (*Liu et al., 2015*; *Oda, 1984*; *Wokke et al., 1990*; *Banker et al., 1983*). Denervation with subsequent reinnervation can lead to degenerated NMJ morphology. For instance we previously demonstrated, in adult mice, sciatic nerve disruption-mediated denervation at a considerable distance from the target lower limbs and subsequent reinnervation leads to a high fraction of NMJs with degenerated morphology (*Liu et al., 2015*). In contrast, insults in relatively close proximity to nerve terminals that lead to reinnervation, even in aged skeletal muscle, result in efficient repair without degenerated morphology (*Kang and Lichtman, 2013*). Although loss of NMJ-organizers such as Agrin or Laminin-α4 is sufficient to induce denervation, these molecules are not lost with age (*Samuel et al., 2012*). Furthermore, denervation stemming from sciatic nerve injury in adults or through genetic disruption of critical NMJ regulators in the embryo leads to robust elevation or induction of extra-synaptic expression of synapse-enriched genes (*Figure 1—figure supplement 2*) (*Liu et al., 2015*; *Sanes and Lichtman, 1999*). In aged skeletal muscles we did not observe characteristics of robust denervation, therefore denervation is likely not a principal mechanism driving age-related NMJ degeneration.

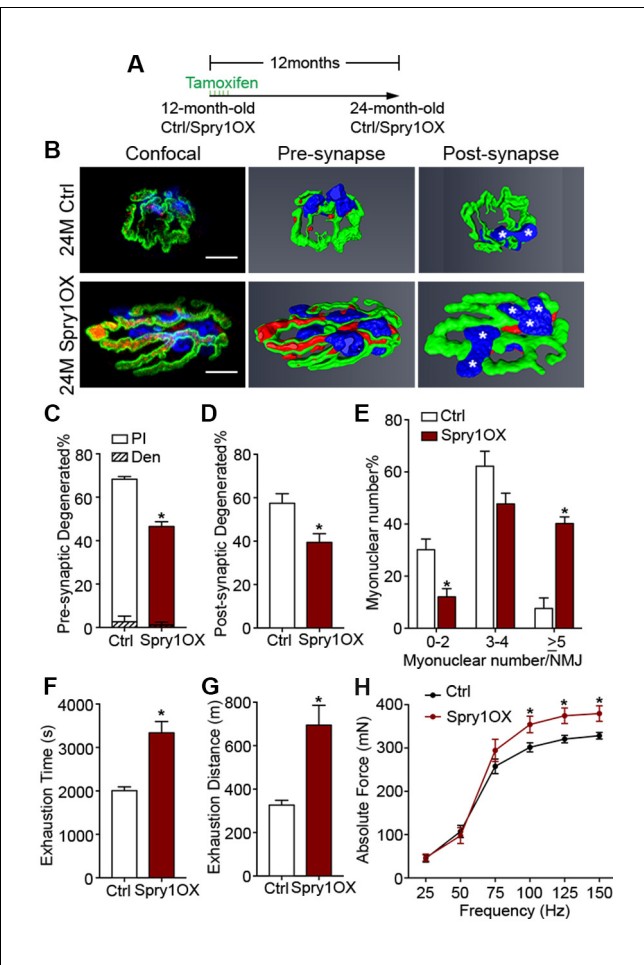

**Figure 4.** Prevention of age-related SC loss by *Spry1* overexpression is associated with attenuation of sarcopenia-related declines in NMJ and skeletal muscle integrity. (**A**) Scheme demonstrating time of tamoxifen treatment and harvest of tissue for Ctrl or Spry1OX mice. (**B**) Confocal IF images and 3-D Amira-based reconstructions (viewed from the pre-synaptic or post-synaptic side) of aged Ctrl and Spry1OX NMJs stained for post-synaptic AChRs (green), nerve terminal markers (red) and nuclei (blue). Post-synaptic myonuclei are indicated with white asterisks. Scale bar = 10 μm. (**C–D**) Quantification of (**C**) pre-synaptic degenerated, including partially innervated (PI) or completely denervated (Den), or (**D**) post-synaptic degenerated NMJ percentage of aged Ctrl and Spry1OX TA muscles. (**E**) Percentage distribution of TA NMJs based on number of post-synaptic myonuclei. (**F–G**) Treadmill performance of aged Ctrl and Spry1OX mice, measured as (**F**) exhaustion time and (**G**) exhaustion distance. (**H**) Peak absolute force generated by aged Ctrl and Spry1OX EDL muscles stimulated at different frequencies ex vivo. *$p < 0.05$ compared to corresponding Ctrl, unpaired Student's *t*-test. $n = 4$ mice per group, >20 NMJs/mouse (**C–E**); $n = 3$ (Ctrl) or 5 (Spry1OX) mice (**F–G**); $n = 4$ (Ctrl) or 6 (Spry1OX) mice (**H**).

The following figure supplements are available for figure 4:

**Figure supplement 1.** Forced expression of Spry1 in SCs attenuates age-associated SC loss.

**Figure supplement 2.** Forced expression of Spry1 in SCs impedes myofiber atrophy.

**Figure supplement 3.** Model of the interrelationship between SC loss and NMJ deterioration with age.

Recently SCs have been shown to contribute to sedentary aging myofibers (*Keefe et al., 2015*). Through the use of a distinct SC fate tracking strategy and single myofiber analysis at 6, 12, 18, and 24 months, we were able to address at which ages SC-derived contributions near NMJs prevail and when they decline. Such contributions could reflect severe myofiber degeneration and regeneration,

as observed with myofiber insult (*Li et al., 2011*; *Li and Thompson, 2011*). Although laser ablation adjacent to endplate regions leads to local myofiber degeneration and regeneration with NMJ fragmentation, whether this insult affected the number, function, or recruitment of SCs or other skeletal muscle resident stem and progenitor cell types is unknown (*Li and Thompson, 2011*; *Bentzinger et al., 2013*; *Judson et al., 2013*). Central nucleated myofibers, an indicator of degeneration and regeneration, have been observed in aged muscles (*Valdez et al., 2010*). Moreover, degenerative-regenerative events have been observed along the length of aging myofibers; however, such events leading to degenerated NMJ morphology in aged muscles are rare (*Li et al., 2011*). In contrast, we observed SC derived progenitor fusion into myofibers without appreciable central nucleation, consistent with previous reports during aging, adolescent maturation, and denervation atrophy (*Liu et al., 2015*; *Keefe et al., 2015*; *Pawlikowski et al., 2015*). During natural aging or upon SC depletion, loss of SC contribution is associated with NMJ degeneration. Correspondingly, we found correlations between indices of NMJ degeneration and reduced post-synaptic myonuclear cluster size. Post-synaptic myonuclear cluster size can be influenced by dissociation through the manipulation of proteins that anchor nuclei (*Zhang et al., 2007*; *Grady et al., 2005*). Aging is associated with loss of total number and normal spatial arrangement of myonuclei (*Bruusgaard et al., 2006*; *Brack et al., 2005*). Therefore, it is likely that both loss and dissociation contribute to reductions in the size of post-synaptic myonuclear clusters with age.

Stimulation of Spry1 expression has been shown to prevent age-related SC loss (*Chakkalakal et al., 2012*; *Bigot et al., 2015*). In primary cultures of aged human myoblasts, stimulation of Spry1 expression through global epigenetic modification promoted SC maintenance at the expense of myogenic differentiation (*Bigot et al., 2015*). Similarly, forced expression of the related Spry2 during embryonic myogenesis favored retention of renewal competent as opposed to terminally committed myogenic cells (*Lagha et al., 2008*). In contrast, based on restoration of post-synaptic myonuclei, prevention of age-related atrophy, and improved force generation, we demonstrate that SC-specific Spry1 overexpression attenuates age-related SC decline without compromising myogenic differentiation. In the context of aging, Spry1 modifies FGF activity to regulate SC pool size at homeostasis; however, the mechanisms of Spry1-mediated regulation of adult and aged post-synaptic myogenic differentiation are unknown (*Chakkalakal et al., 2012*). Furthermore, fusion of SC derived Spry1OX myogenic progenitors could lead to suppression of extracellular signal-regulated kinase (ERK) activity, which is elevated in skeletal muscle biopsies from chronic obstructive pulmonary disorder (COPD) patients and sedentary aged individuals (*Lemire et al., 2012*; *Williamson et al., 2003*). Although it is unclear which resident cell(s) within COPD or aged biopsies possessed the heightened ERK activity, it would be of interest to determine sarcopenia outcomes upon myofiber-specific Spry1 overexpression. Regardless, in addition to maintaining skeletal muscle regenerative potential, Spry1 manipulation provides a viable approach for counteracting sarcopenia-related NMJ deterioration and aged muscle function decline. Collectively, our observations, obtained through multiple lines of query, demonstrate that loss of myofiber regenerative potential endowed in SCs is a significant mediator of age-related NMJ degeneration.

## Materials and methods

### Animals

Wild type C57BL/6J mice were obtained from National Institute on Aging; $Pax7^{CreERT2}$ (017763), $Rosa26^{mTmG}$ (007576), $Rosa26^{nTnG}$ (023035) and $Rosa26^{DTA}$ (009669) mice were obtained from Jackson Laboratories (Bar Harbor, Maine); mice carrying a transgene encoding a Cre-inducible expression construct for *Spry1* controlled by a chicken *β*-actin gene (CAG) promoter were used for Spry1-overexpression studies (*Chakkalakal et al., 2012*; *Yang et al., 2008*). $Rosa26^{mTmG}$, $Rosa26^{nTnG}$, $Rosa26^{DTA}$ and $CAG^{Spry1}$ mice were crossed with $Pax7^{CreERT2}$ mice to generate $Pax7^{CreER/+}$; $Rosa26^{mTmG/+}$ (P7mTmG), $Pax7^{CreER/+}$; $Rosa26^{nTnG/+}$ (P7nTnG), $Pax7^{CreER/+}$; $Rosa26^{DTA/+}$ (P7DTA), $Pax7^{CreER/+}$; $CAG^{Spry1/+}$ (Spry1OX) mice and control CreER negative (Ctrl) littermates. Tamoxifen (Sigama-Aldrich, T5648, St. Louis, Missouri) was administrated at a dose of 2.0 mg/day for five consecutive days to induce Cre recombination at given ages. All animal procedures were conducted in accordance with institutional guidelines approved by the University Committee on Animal Recourses, University of Rochester Medical Center.

## Immunofluorescence (IF)

Muscles were incubated at 4°C overnight in 30% sucrose solution, embedded in OCT compound (Tissue Tek, Torrance, California) and frozen in dry-ice-cooled isopentane. Flash-frozen muscles were sectioned at 10 μm (for transverse sections) or 30 μm (for longitudinal sections) and stored at −80°C. Sections or cultured cells were fixed for 3 min in 4% paraformaldehyde (PFA) (no PFA fixation for MyHC antibodies), incubated with 0.2% Triton X-100 for 10 min, blocked in 10% normal goat serum (NGS, Jackson ImmunoResearch, West Grove, Pennsylvania) for 30 min at room temperature and stained with primary antibodies. If necessary (when mouse primary antibodies were used), sections were blocked in 3% affinipure Fab fragment goat anti-mouse IgG(H+L) (Jackson ImmunoResearch, 115-007-003) with 2% NGS at room temperature for 1 hr. Primary antibodies were incubated at 4°C overnight or 2 hr at room temperature, and secondary antibodies were incubated for 1 hr at room temperature. For whole-mounts, diaphragms were fixed in 2% PFA for 30 min on ice and blocked in 0.5% Triton X-100 with 10% NGS for 1 hr at room temperature; primary antibodies were incubated at 4°C for 24 hr and secondary antibodies were incubated for 2 hr at room temperature. DAPI (Invitrogen) staining was used to mark nuclei. All slides were mounted with Fluoromount-G (SouthernBiotech, Birmingham, Alabama).

## Conventional and confocal IF microscopy and analysis

Transverse sections and cells were imaged on a Zeiss Axio Observer A.1 microscope. Longitudinal sections were stained with fluorescently conjugated α-bungarotoxin (BTX) and a mixture of antibodies (synaptic vesicle protein 2, SV2; synaptotagmin-2, Syt-2; and neurofilament) to label acetylcholine receptors (AChRs) and nerve terminals respectively. Sections were viewed with an Olympus Fluoview 1000 confocal microscope with 60X objective at a 0.44 μm step size. Amira software was used to generate 3-D reconstructed NMJs for innervation analysis, identifying and enumerating post-synaptic myonuclei, and quantifying AChR expressing volume and intensity. Max-projection z-stack images of NMJs were generated with ImageJ software. The post-synaptic side was identified based on the entry of the terminal axon and as the concave side of the NMJ. We considered an NMJ to be: (i) innervated, if the vast majority of post-synaptic regions are covered by pre-synaptic terminal markers; (ii) pre-synaptic degenerated, if >5 μm length of an AChR enriched branch within the post-synaptic apparatus is not covered by pre-synaptic terminal markers (partially innervated), or if >90% of the post-synaptic apparatus is devoid of pre-synaptic nerve terminal markers (completely denervated); (iii) post-synaptic degenerated, if the AChR-enriched area resembled a patch devoid of defined elaborate branches >5 μm in length (*Liu et al., 2015*). Post-synaptic myonuclei are identified based on 4′,6-Diamidino-2-Phenylindole (DAPI) labeling (>25% DAPI covered by the BTX+ post-synaptic apparatus) (*Liu et al., 2015*; *Zhang et al., 2007*).

## RNA isolation and RT-qPCR

To separate the synaptic and extra-synaptic regions, diaphragms were incubated in ice-cold diethyl pyrocarbonate (DEPC)-treated PBS with BTX for 30 min, and dissected based on NMJ staining under a conventional IF microscope. Muscles were submerged in Trizol (Ambion, Carlsbad, California) and homogenized using Bullet Blender Gold (Nextadvance BB24-AU, Averill Park, New York). RNA was extracted with the RNeasy Mini Kit (Qiagen, Hilden, Germany) according to the manufacturer's instructions and prepared for RT-qPCR analysis or RNA-sequencing experiments. First-strand complementary DNA was synthesized from 200 ng of RNA using the SuperScript First-Strand cDNA Synthesis Kit (Invitrogen, Carlsbad, California). RT-qPCR was performed on a Step One Plus Real Time PCR machine (Applied Biosystems, Foster City, California), with Platinum SYBR Green qPCR Super-Mix-UDG and ROX master mix (Invitrogen) using primers against AChE, Chrna1, Chrnd, Chrne, Etv5, Musk and B2M (*Supplementary file 1*). All reactions for RT-qPCR were performed using the following thermal cycler conditions: 50°C for 2 min, 95°C for 2 min, 40 cycles of a two-step reaction, denaturation at 95°C for 15 s, annealing at 60°C for 30 s. Variant expression was normalized to B2M using the comparative $C_t$ method and reported relative to the average of the synaptic samples.

## RNA-sequencing

RNA concentration was determined with the NanoDrop 1000 spectrophotometer (NanoDrop, Wilmington, Delaware) and RNA quality was assessed with the Agilent Bioanalyzer (Agilent

Technologies, Santa Clara, California). Illumina-compatible library construction was performed using the TruSeq Stranded Total RNA Sample Preparation Kit (Illumina, San Diego, California) per manufacturer's protocols. The amplified libraries were hybridized to the Illumina single-end flow cell and amplified using the cBot (Illumina) at a concentration of 8 pM per lane. Single-end reads of 100 nt were generated for each sample. Sequenced reads were cleaned according to a rigorous pre-processing workflow (Trimmomatic-0.32) before mapping some of them to the mouse reference genome (GRCm38.p4) with STAR-2.4.2a. Cufflinks2.0.2 with the gencode-M6 mouse gene annotations was then used to perform differential expression analysis.

## Gene set enrichment analysis

The gene expression dataset was transferred to the Gene Set Enrichment Analysis software (http://www.broadinstitute.org/gsea) and the analysis was carried out with default parameters except that gene permutation was applied. Gene sets derived from the Molecular Function Ontology (c5.mf.v5.2.symbols) (http://www.broadinstitute.org/gsea/msigdb) were utilized to determine synaptic gene enrichment. The enrichment results were visualized using the Cytoscape plug-in enrichment map.

## Fluorescence-activated cell sorting (FACS) and cell culture

To obtain purified SCs, primary cells were isolated as described previously (*Paris et al., 2016*). Muscles were dissected from mice and digested using collagenase II (Gibco, Carlsbad, California) in Dulbecco's modified Eagles medium (DMEM, Sigma-Aldrich) for 60 min at 37°C with agitation. The suspension was then washed and digested in collagenase II and dispase (Gibco) for 30 min at 37°C with agitation. The resultant mononuclear cells were then stained with the following antibodies: CD31-PECy7 (BD Biosciences 561410, San Jose, California), CD45-PECy7 (BD Biosciences 552848), Sca1-FITC (BD Biosciences 562058), Integrin α7 (ITGA7)-Alexa Fluor 647 (AbLab, Vancouver, Canada) and VCAM-PE (Invitrogen RMCD10604). FACS was performed using a FACSAria II Cell Sorter (BD Biosciences) and SCs were collected according to the following sorting criteria: CD31-/CD45-/Sca1-/ITGA7+/VCAM+. FACs-purified SCs were plated at 3000 cells per well in eight-well Permanox chamber slides (Nunc Lab-Tek, Carlsbad, California) and cultured for five days in DMEM with 10% Horse Serum (Thermo Fisher Scientific, Carlsbad, California) and 5 ng/mL FGF2 (Cell Signaling Technologies, Danvers, Massachusetts).

## Single myofiber analysis

For GFP localization, myofibers were purified by conventional collagenase digestion and trituration with fire-polished glass pipets as previously described (*Liu et al., 2015*; *Zammit et al., 2004*). Briefly, the EDL muscle was dissected, rinsed, and incubated in DMEM with 0.1% type I collagenase and 0.1% type II collagenase at 37°C for 60 min. Following digestion, single myofibers were released by gently triturating the EDL with fire-polished-tip Pasteur pipettes. Purified single myofibers were fixed with 4% PFA for 3 min followed by GFP and BTX IF. For assessing single myofiber size, lower limbs were fixed in 4% PFA for 48 hr prior to muscle dissection. Fixed muscles were incubated in 40% NaOH for 2 hr, and single myofibers were dissociated by gently pipetting up and down. Released fibers were then washed in PBS and processed for DAPI staining (*Liu et al., 2015*; *Brack et al., 2005*).

## Quantification of myofiber type composition

Myofiber type composition was quantified using a newly developed ImageJ plug-in (available at https://github.com/aidistan/ij-fibersize; with a copy archived at https://github.com/elifesciences-publications/ij-fibersize) that identifies myofiber types by MyHC isoform expression combined detecting myofiber boundary with Laminin IF (*Tan, 2016*).

## Ex vivo muscle force generation assay

Muscle force generation capacity was analyzed in EDLs using an ASI muscle contraction system (Aurora Scientific, Aurora, Canada) as described previously (*Wei-Lapierre et al., 2013*). Briefly, mice were anaesthetized and TA muscles removed. EDLs were then carefully isolated, adjusted to optimal length ($L_o$) and then stimulated at different frequencies to obtain absolute force values. Muscle force

was recorded and analyzed using Dynamic Muscle Control and Graph Pad Prism software. EDL muscle cross-sectional area was calculated using the equation of cross-sectional area = (muscle mass)/[muscle density (1.06 g/cm$^3$) × optimal fiber length (0.44 × $L_o$)] (*Hakim et al., 2011*).

## Treadmill test

The mice were trained to get used to the treadmill (Columbus Instruments) for three days: first and second days - 5 m/min for 5 min, third day - 7 m/min for 5 min. The exhaustion test is as followed: 5 m/min for 5 min; 10 m/min for 25 mins; 15 m/min for 20 mins; 20 m/min for 15 mins; 21 m/min for 1 min; 22 m/min for 1 min; 23 m/min for 1 min; 24 m/min for 1 min; 25 m/min infinite until the mice cannot run anymore. Continued running was encouraged by delivering brief spirits of air on the mouse's backside using a Whoosh Duster. Exhaustion was defined as the inability of the animal to remain on the treadmill despite the air spray. Both the training and the test were performed without any inclines.

## Antibodies

Pax7 (mouse IgG1, 1:100, Developmental Studies Hybridoma Bank (DSHB), Iowa City, Iowa), Laminin (rabbit, 1:1500, Sigma-Aldrich L9393), GFP (rabbit, 1:400, Millipore AB3030P, Billerica, Massachusetts), SC-71 (MyHC-IIA, mouse IgG1, 1:40, DSHB), BF-F3 (MyHC-IIB, mouse IgG1, 1:40, DSHB), SV2 (mouse IgG1, 1:100, DSHB), Znp-1 (Syt-2, mouse IgG1, 1:200, DSHB) and 2H3 (neurofilament, mouse IgG1, 1:200, DSHB), AlexaFluor 488-conjugated α-Bungarotoxin (1:1000, Thermo Fisher Scientific B-13422), AlexaFluor 647-conjugated α-Bungarotoxin (1:1000, Thermo Fisher Scientific B-35450), AlexaFluor 594-conjugated goat anti-mouse IgG (1:1500, Thermo Fisher Scientific A-11032), AlexaFluor 594-conjugated goat anti-mouse IgG1 (1:1500, Thermo Fisher Scientific A-21125), AlexaFluor 488-conjugated goat anti-mouse IgM (1:1500, Thermo Fisher Scientific A-21042), AlexaFluor 488-conjugated goat anti-rabbit IgG (1:1500, Thermo Fisher Scientific A-32731), AlexaFluor 647-conjugated goat anti-rabbit (1:1500, Thermo Fisher Scientific A-21244).

## Statistical analysis

Results are presented as mean + s.e.m. Statistical significance was determined by Student's *t*-tests for simple comparison, two-way ANOVA/Sidak for multiple comparisons with Graph Pad Prism software. $p < 0.05$ was considered as statistically significant.

## Acknowledgements

We thank Dr. Gabrielle Kardon (University of Utah) for providing the *Pax7$^{CreER}$* mouse line to Jackson Laboratories; Dr. Robert Friesel (Maine Medical Center Research Institute) for kindly providing the *CAG$^{Spry1}$* mice; Dr. Paivi Jordan (University of Rochester Medical Center Confocal and Conventional Microscopy Core), Dr. Jonathan Carroll-Nellenback (University of Rochester Center for Integrated Research Computing), University of Rochester Medical Center Flow Cytometry Core, University of Rochester Genomics Research Center and Center for Musculoskeletal Research Histology, Biochemistry, and Molecular Imaging Core for technical assistance. This work was supported by URMC start-up funds, Dept. of Defense grant (W81XWH-14-1-0454) and NIH grant (RO1AG051456) to JVC; NYSTEM award (C026877) to WL.

## Additional information

### Funding

| Funder | Grant reference number | Author |
| --- | --- | --- |
| National Institute on Aging | RO1AG051456 | Joe V Chakkalakal |
| Congressionally Directed Medical Research Programs | W81XWH-14-1-0454 | Joe V Chakkalakal |
| New York Stem Cell Foundation | C026877 | Wenxuan Liu |

The funders had no role in study design, data collection and interpretation, or the decision to submit the work for publication.

## Author contributions

WL, Conceptualization, Data curation, Formal analysis, Writing—original draft, Writing—review and editing; AK, NDP, Data curation, Formal analysis, Validation, Investigation, Methodology; SF, Data curation, Formal analysis, Validation, Investigation; LW-L, RTD, Formal analysis, Supervision, Methodology; MC-L, Data curation, Formal analysis, Methodology; AT, Data curation, Software, Methodology; MF, Data curation, Formal analysis, Validation; PM, Software, Supervision, Investigation, Methodology; JVC, Conceptualization, Supervision, Funding acquisition, Writing—original draft, Project administration, Writing—review and editing

## Author ORCIDs

Nicole D Paris, http://orcid.org/0000-0003-0654-0983
Joe V Chakkalakal, http://orcid.org/0000-0002-8440-7312

## Ethics

Animal experimentation: Animal experimentation: This study was performed in strict accordance with the recommendations in the Guide for the Care and Use of Laboratory Animals of the National Institutes of Health. Work with mice was conducted in accordance with protocols approved by the University Committee on Animal Resources, University of Rochester Medical Center protocol (#101565/2013-002)

## Additional files

### Supplementary files

• Supplementary file 1. RT-qPCR primer list.

### Major datasets

The following dataset was generated:

| Author(s) | Year | Dataset title | Dataset URL | Database, license, and accessibility information |
|---|---|---|---|---|
| Wenxuan Liu, Joe V Chakkalakal | 2017 | Adult/aged diaphragm and gastrocnemius RNA seq | https://www.ncbi.nlm.nih.gov/Traces/study/?acc=SRP107891 | Publicly available at the NCBIShort Read Archive (accession no: SRP107891) |

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
