## [Decision Letter]

Thank you for submitting your article "Loss of adult skeletal muscle stem cells drives age-related neuromuscular junction degeneration" for consideration by *eLife*. Your article has been favorably evaluated by Sean Morrison (Senior Editor) and three reviewers, one of whom, Amy J Wagers (Reviewer #3), is a member of our Board of Reviewing Editors. The following individuals involved in review of your submission have agreed to reveal their identity: Gregorio Valdez (Reviewer #1) and Colin Crist (Reviewer #2).

The reviewers have discussed the reviews with one another and the Reviewing Editor has drafted this decision to help you prepare a revised submission.

Summary:

In a previous study published in *eLife* in 2015, this group used elegant genetic tools to inducibly label or deplete muscle stem cells (MuSCs). By doing so, they showed that MuSC depletion leads to an increase in muscle atrophy, fibre type transitions and fibrosis, with a corresponding decrease in the generation of muscle force. Furthermore, MuSC depletion followed by denervation resulted in decreased reinnervation, decreased myonuclei at neuromuscular junctions (NMJs) and poor post-synaptic morphology. MuSC labeling experiments demonstrated that MuSCs contribute directly to NMJs. This Research Advance submission builds on this previous work using similar genetic strategies to investigate satellite cell incorporation into myofibers as a function of age and its effect on neuromuscular junction (NMJ) integrity. The authors report reduced integrity of NMJs with advancing age and use an mT/mG fluorescence tracing system (described in their earlier *eLife* publication) to correlate the presence of these degenerated junctions with reduced incorporation of satellite cells in the region of the NMJ. Finally, they show using a diphtheria toxin ablation system to deplete muscle satellite cells endogenously that loss of satellite cells causes precocious loss of NMJ integrity and that overexpression of Spry1 using a Pax7CreER driven induction system protects from age-associated NMJ degeneration and promotes maintenance of muscle mass and strength. Altogether, the work provides a number of novel and important observations relevant for understanding the detrimental effects of aging on skeletal muscle and physical performance and highlights a previously unappreciated role for satellite cells in these effects. There are, however, a few remaining questions about some of the authors' conclusions that need to be addressed:

Essential revisions:

1) The manuscript lacks data showing directly that postsynaptic myonuclei originate from muscle stem cells. The most relevant data for this idea is presented in Figure 3; however, it is impossible to conclude that postsynaptic nuclei emanate from satellite cells based on these data alone. The presence of GFP (Figure 3) only proves that satellite cells are present throughout a given muscle fiber, including proximal to the postsynaptic region. Additional approaches (e.g. generation of mice with nuclear rather than membrane GFP, or tagging of satellite cells with a nuclear marker ex vivo and injection into the synaptic region to see where their nuclei end up in already formed muscle fibers) would be needed to prove that a postsynaptic myonuclei emanate from satellite cells and thus are lost following depletion of this cell population. If such data are available, they should be added to the manuscript. If not, then the authors should soften their conclusion as stated in the Discussion – "Specifically, SCs are a source of post-synaptic myonuclei" of matured muscle fibers, as there is simply no data directly showing this.

2) The authors state that little denervation ("no gross denervation") occurs in aged animals. But in Figure 1, the majority of NMJs are either completely or partially denervated, and this data is represented as partially missing nerve (Inn). These findings appear to support published data showing that most aged NMJs are partially denervated and a smaller population completely denervated. While the authors argue that their supplemental data (Figure 1—figure supplement 2) that the transcriptional program of myonuclei in aged muscle differs from that seen at a single time point after sciatic nerve transection further supports a lack of denervation in aged muscle, it is unclear how this time point was chosen and what sort of overlap one might expect that could have led to the opposite conclusion, given that the sciatic nerve transection model is obviously a much more severe, and also very different, surgical model. Thus, the conclusion that there is limited denervation in aged muscle is not particularly compelling, and the authors are advised to change the "Inn" label in this Figure 1 to partially innervated (PI) or partially denervated (PD).

3) Since the loss of myonuclei is proposed to cause aging of NMJs, it is important to show that myonuclei disappear prior to aging of NMJs. However, there is no data supporting that this sequence of events occurs during aging. Instead, it is shown that a similar number of nuclei are present at NMJs of 12 compared to 6 months of age, see Figure 3. Since perisynaptic Schwann cells are also affected by aging, it is important to determine their contribution to NMJ-associated nuclei in young and old muscles. Answer to these questions would better establish the relationship between aging of NMJs and loss of myonuclei specifically derived from stem cells.

4) Figure 3 suggests reduced incorporation of new satellite cells overall (reduced total number of mGFP+ fibers) with aging. This is a bit counterintuitive, however, given the increased frequency of centrally nucleated fibers normally detected in aged muscle (e.g., Valdez et al. 2010). Can the authors explain/ comment on this?

5) It is plausible that NMJ-associated myonuclei dissociate from the postsynaptic region in atrophying muscle fibers. This possibility should be explored or discussed.

6) Incorporation of satellite cells into myofibers in the Spry1OX system will lead to Spry1 overexpression in differentiated fibers as well. Thus, some of the effects the authors see with respect to protection from age-related NMJ dysfunction in this system may relate to Spry1 functions in fibers. It is therefore important to evaluate the level of transgene expression in muscle fibers over time, and ideally this issue should be addressed experimentally by analysis of an overexpression system in which fiber-specific Spry1 overexpression is induced, or possibly by transplantation of WT satellite cells not their existing Spry1OX system (though this may be technically challenging). This issue is particularly relevant as Spry1 has been reported as a negative regulator of ERK signaling and ERK activation has been associated with muscle atrophy in certain settings (e.g., Lemire et al., 2012). Thus, it is possible that introduction of Spry1 into myofibers via satellite cell fusion in the Spry1OX system protects them from subsequent age-related atrophy. If the authors are unable to provide direct experimental evidence addressing this point, they must then provide an explicit discussion of this possibility and revise their conclusions accordingly.

---

## [Author Response]

*Essential revisions:*

*1) The manuscript lacks data showing directly that postsynaptic myonuclei originate from muscle stem cells. The most relevant data for this idea is presented in Figure 3; however, it is impossible to conclude that postsynaptic nuclei emanate from satellite cells based on these data alone. The presence of GFP (Figure 3) only proves that satellite cells are present throughout a given muscle fiber, including proximal to the postsynaptic region. Additional approaches (e.g. generation of mice with nuclear rather than membrane GFP, or tagging of satellite cells with a nuclear marker ex vivo and injection into the synaptic region to see where their nuclei end up in already formed muscle fibers) would be needed to prove that a postsynaptic myonuclei emanate from satellite cells and thus are lost following depletion of this cell population. If such data are available, they should be added to the manuscript. If not, then the authors should soften their conclusion as stated in the Discussion – "Specifically, SCs are a source of post-synaptic myonuclei" of matured muscle fibers, as there is simply no data directly showing this.*

We have generated Pax7CreER; Rosa26nTnG (P7nTnG) mice, which indelibly label SCs upon tamoxifen administration with a nuclear-GFP (nGFP). At 4.5 months P7nTnG mice were injected with tamoxifen and EDL single myofibers were collected immediately or 6 weeks thereafter. Immediately after tamoxifen administration only Pax7+ SCs were nGFP labeled. Through analysis of NMJs 6 weeks after tamoxifen administration nGFP+ post-synaptic myonuclei were observed. These results are included as new Figure 3—figure supplement 1, and accompanying text included in the Methods (subsection “Animals”) and Results (subsection “SCs contribute to adult and middle-aged, but not aged NMJs”, last paragraph).

*2) The authors state that little denervation (" no gross denervation") occurs in aged animals. But in Figure 1, the majority of NMJs are either completely or partially denervated, and this data is represented as partially missing nerve (Inn). These findings appear to support published data showing that most aged NMJs are partially denervated and a smaller population completely denervated. While the authors argue that their supplemental data (Figure 1—figure supplement 2) that the transcriptional program of myonuclei in aged muscle differs from that seen at a single time point after sciatic nerve transection further supports a lack of denervation in aged muscle, it is unclear how this time point was chosen and what sort of overlap one might expect that could have led to the opposite conclusion, given that the sciatic nerve transection model is obviously a much more severe, and also very different, surgical model. Thus, the conclusion that there is limited denervation in aged muscle is not particularly compelling, and the authors are advised to change the "Inn" label in this Figure 1 to partially innervated (PI) or partially denervated (PD).*

We have modified graphs in Figure 1, Figure 3 and Figure 4 and the relevant text to include complete denervation (Den) and partial innervation (PI) designations. We agree sciatic nerve injuries are more severe than age-related muscle decline; however, such models have been used to study muscle aging (Xu et al., J Cachexia Sarcopenia Muscle, 2017). The denervation/reinnervation regimen for Figure 1—figure supplement 2 was chosen based on the Liu et al., *eLife* 2015 study where it was demonstrated that such a model does lead to full or partial innervation with degenerated NMJ morphology akin to what is observed in aged muscle. The overlap between our denervation/reinnervation regimen and aging is the persistence of degenerated NMJ morphology. We have included statements for the above where Figure 1—figure supplement 2 is reported (subsection “Complete denervation is not a prominent feature of aged skeletal muscles”).

*3) Since the loss of myonuclei is proposed to cause aging of NMJs, it is important to show that myonuclei disappear prior to aging of NMJs. However, there is no data supporting that this sequence of events occurs during aging. Instead, it is shown that a similar number of nuclei are present at NMJs of 12 compared to 6 months of age, see Figure 3. Since perisynaptic Schwann cells are also affected by aging, it is important to determine their contribution to NMJ-associated nuclei in young and old muscles. Answer to these questions would better establish the relationship between aging of NMJs and loss of myonuclei specifically derived from stem cells.*

The data presented in Figure 3 are conducted with ANOVA. As requested to compare 6 and 12 month-old Ctrl NMJs we have conducted pair-wise t-test. Pair-wise comparison demonstrates that at 12 months there is a significant increase in the proportion of NMJs with 0-2 post-synaptic myonuclei, or post-synaptic degenerated NMJs. In contrast we there was no significant difference in pre-synaptic degenerated (i.e. partially innervated) NMJs at 12 compared to 6 months. We have now included these results as Figure 3—figure supplement 3.

It is well established that Schwann cells are on the nerve terminal side and as far as we know do not contribute to post-synaptic myonuclei or have myogenic potential (subsection “Complete denervation is not a prominent feature of aged skeletal muscles”, last paragraph). Conversely, we have not observed GFP labeled pre-synaptic nuclei in P7mTmG or P7nTnG mice during aging (this study) or denervation/reinnervation muscle atrophy (Liu et al., *eLife*, 2015). Since satellite cells are the principal source of myonuclei during repair or regeneration we focused on post-synaptic myonuclear number.

*4) Figure 3 suggests reduced incorporation of new satellite cells overall (reduced total number of mGFP+ fibers) with aging. This is a bit counterintuitive, however, given the increased frequency of centrally nucleated fibers normally detected in aged muscle (e.g., Valdez et al. 2010). Can the authors explain/ comment on this?*

As requested we have commented on satellite cell derived contribution in relation to central nucleation, see the last paragraph of the Discussion. Central nucleation is often used as an indicator of a myofiber that has at some point undergone a major bout of degeneration and regeneration. Therefore, Figure 6 of the Valdez et al. 2010 study would indicate that at some point between 5 and 23-24 months a small fraction of myofibers underwent a severe bout of degeneration and regeneration, the cause and timing of which is unknown. Regardless, SC derived progenitor fusion into myofibers has been observed without appreciable central nucleation, see Keefe et al., Nat Comm, 2015; Liu et al., *eLife*, 2015; Pawlikowski et al., Skeletal Muscle, 2015.

*5) It is plausible that NMJ-associated myonuclei dissociate from the postsynaptic region in atrophying muscle fibers. This possibility should be explored or discussed.*

We agree it is possible that post-synaptic myonuclei may have been displaced with age (Bruusgaard et al., J Appl Physiol, 2006), however given the overall declines in myonuclei number starting at 12 months (Brack et al., J Cell Sci, 2005) it is likely that both loss and displacement contribute. As suggested, we have included a discussion of the above (Discussion, third paragraph).

*6) Incorporation of satellite cells into myofibers in the Spry1OX system will lead to Spry1 overexpression in differentiated fibers as well. Thus, some of the effects the authors see with respect to protection from age-related NMJ dysfunction in this system may relate to Spry1 functions in fibers. It is therefore important to evaluate the level of transgene expression in muscle fibers over time, and ideally this issue should be addressed experimentally by analysis of an overexpression system in which fiber-specific Spry1 overexpression is induced, or possibly by transplantation of WT satellite cells not their existing Spry1OX system (though this may be technically challenging). This issue is particularly relevant as Spry1 has been reported as a negative regulator of ERK signaling and ERK activation has been associated with muscle atrophy in certain settings (e.g., Lemire et al., 2012). Thus, it is possible that introduction of Spry1 into myofibers via satellite cell fusion in the Spry1OX system protects them from subsequent age-related atrophy. If the authors are unable to provide direct experimental evidence addressing this point, they must then provide an explicit discussion of this possibility and revise their conclusions accordingly.*

Similar to the Lemire et al. study in muscles of chronic obstructive pulmonary patients, elevated p38 MAPK, ERK1/2, and *JNK* have been reported in sedentary aged muscles (Williamson et al., J Physiol, 2003). In either study it is not clear which cell(s) (i.e. myofibers, SCs, interstitial cells, blood vessels, etc.) contained the elevated activities. Unfortunately, we do not have the myofiber specific Spry1OX system. Accordingly, we have provided more discussion to include the possibility of SC-derived Spry1OX myofiber activity as an alternative (subsection “Animals”). In addition, we have also modified our statements and conclusions to account for this possibility (Results subsection “Aged NMJ and skeletal muscle integrity is improved upon Spry1 overexpression in SCs”, Discussion, second paragraph, subsection “Animals” and Figure 4 legend title).